# Investigating the Prognostic Relevance of Tumor Immune Microenvironment and Immune Gene Assembly in Breast Carcinoma Subtypes

**DOI:** 10.3390/cancers14081942

**Published:** 2022-04-12

**Authors:** Beáta Szeitz, Orsolya Pipek, Janina Kulka, Csilla Szundi, Orsolya Rusz, Tímea Tőkés, Attila Marcell Szász, Kristóf Attila Kovács, Adrián Pesti, Taya Beri Ben Arie, Ambrus Gángó, Zsolt Fülöp, Emőke Drágus, Stefan A. Vári-Kakas, Anna Mária Tőkés

**Affiliations:** 1Division of Oncology, Department of Internal Medicine and Oncology, Semmelweis University, 1083 Budapest, Hungary; szeitz.beata@phd.semmelweis.hu (B.S.); timi.tokes@gmail.com (T.T.); szaszam@gmail.com (A.M.S.); 2Department of Physics of Complex Systems, Institute of Physics, Eötvös Loránd University, 1117 Budapest, Hungary; orsolya.pipek@ttk.elte.hu; 3Department of Pathology, Forensic and Insurance Medicine, Semmelweis University, 1091 Budapest, Hungary; janinakulka@gmail.com (J.K.); szundi.csilla@gmail.com (C.S.); attila.drkovacs@gmail.com (K.A.K.); pesti.adrian@med.semmelweis-univ.hu (A.P.); tamiberi@gmail.com (T.B.B.A.); 4Department of Pathology, Forensic and Insurance Medicine, SE NAP, Brain Metastasis Research Group, Semmelweis University, 1091 Budapest, Hungary; masodikanyu@gmail.com; 5Department of Bioinformatics, Semmelweis University, 1052 Budapest, Hungary; 6MTA-SE Momentum Molecular Oncohematology Research Group, 1st Department of Pathology and Experimental Cancer Research, Semmelweis University, 1085 Budapest, Hungary; gangoambrus@gmail.com; 7Department of Surgery, George Emil Palade University of Medicine, Pharmacy, Science and Technology, 540139 Targu Mures, Romania; zsolt_fulop15@yahoo.com; 8Department of Urology, Clinical County Hospital, 540167 Targu Mures, Romania; d.emoke_29@yahoo.com; 9Faculty of Electrical Engineering and Information Technology, University of Oradea, Str. Universitatii nr. 1, 410087 Oradea, Romania; vkstefan@gmail.com

**Keywords:** breast carcinoma subtypes, TIL, immune-related gene, NanoString, CD4^+^ T cells, CD8^+^ T cells

## Abstract

**Simple Summary:**

The tumor immune microenvironment of different breast carcinoma (BC) subtypes and immune gene assembly of metastasizing and non-metastasizing HER2-negative BCs was analyzed. Examination of 309 Hematoxylin and Eosin-stained slides highlighted that the distribution of tumor infiltrating lymphocytes (TILs) from peritumoral, stromal and intratumoral regions varied greatly within all subtypes, with most tumors (66.01%) belonging to the immunologically “cold” group. Hormone receptor (HR) negative subtypes generally showed higher immune activity in all analyzed regions. No survival benefit was detected based on the spatial distribution of TILs. A lower CD4^+^/CD8^+^ ratio at the stromal internal tumor region indicated longer distant metastasis-free survival. When assessing immune gene expression between metastatic and non-metastatic BCs, the list of differentially expressed genes were non-identical across luminal and TNBCs, suggesting that these subtypes may use different mechanisms to bypass the immunological surveillance. Understanding these differences in the immune gene assembly may pave the way to the development of new immune-modulation therapies.

**Abstract:**

We hypothesized that different BC subtypes are characterized by spatially distinct tumor immune microenvironment (TIME) and that immune gene assembly of metastatic (Met) and non-metastatic (Ctrl) BCs vary across subtypes. Peritumoral, stromal and intratumoral TIL was assessed on 309 BC cases. Hot, cold and immune-excluded groups were defined, and the prognostic role of this classification was assessed. CD4^+^/CD8^+^ positivity was analyzed in 75 cases in four systematically predefined tumor regions. Immune gene expression of Met and Ctrl HER2-negative BCs was compared by using NanoString nCounter technology. The amount of TIL infiltration varied greatly within all BC subtypes. Two-third of the cases were cold tumors with no significant survival difference compared to hot tumors. A lower CD4^+^/CD8^+^ ratio at the stromal internal tumor region was significantly associated with longer distant metastasis-free survival. The differentially expressed immune genes between Met and Ctrl varied across the studied BC subtypes with TNBC showing distinct features from the luminal subtypes. The TIME is characterized by a considerable heterogeneity; however, low level of TILs does not equate to disease progression. The differences in immune gene expression observed between Met and Ctrl breast carcinomas call attention to the important role of altered immune function in BC progression.

## 1. Introduction

Over the last few years, important results have been published related to breast cancer immunity: significant knowledge related to immune cell composition, immune response, and mechanisms of immune evasion of cancer has accumulated, but the immunogenicity of different breast carcinoma subtypes and the contribution of the immune microenvironment to the clinical course of metastatic breast cancers are less clear and less extensively studied [1,2,3,4,5,6]. A rapid increase in the number of immunotherapeutic clinical trials can be observed since 2019; the year Atezolizumab was approved for triple negative breast carcinomas. However, there is a need to better understand the immune microenvironment of other breast carcinoma subtypes with the goal of rendering those subtypes also accessible to immunotherapies [7,8,9].

Several aspects of tumor associated immune cells are continuously analyzed and described; thus, different classification systems have been suggested previously. 

Tumor-infiltrating lymphocytes (TILs) are the most widely studied population of tumor-infiltrating immune cells in breast carcinomas and are routinely classified as stromal or intratumoral on Hematoxylin and Eosin (HE)-stained slides based on standardized methodology [4,10,11,12]. Some studies describe the fundamental role of the dynamic interplay between immune cells within the tumor microenvironment and the surrounding tissue on the clinical outcome of breast cancer patients, but recently published guidelines for scoring immune cell infiltrates just partly elaborate on the importance of immune cell localization in different tumor compartments [11,12]. Evidence is accumulating to support the use of TIL scoring as a prognostic biomarker in breast carcinomas and its prognostic benefit is progressively investigated and documented [13,14]. Increased TIL and immune-related gene expression are significantly associated with improved survival in early-stage triple-negative (TNBC), human epidermal growth factor receptor-2 (HER2) positive breast cancer. The high-risk estrogen receptor (ER) positive breast cancers’ group is considered controversial in this aspect [15,16]. The prognostic role of TIL in other hormone receptor (HR) positive breast cancer is more elusive and the interaction between different immune microenvironments and the immune response in this subset of breast carcinomas is less understood [17]. Although, T cells are the predominant TIL population playing a central role in controlling tumor growth, the relationship between quantitative and qualitative differences in T cell subpopulations in different breast carcinoma subtypes and disease prognosis remains disputed [1,14]. The controversies result partly from the fact that TILs are mostly identified based on their phenotypic profiles and not on their functional properties [4]. 

Related to the main components of tumor-infiltrating immune cells isolated from breast cancer CD4^+^ and CD8^+^ T cells are the most abundant immune cells, but B cells also make up a significant proportion of immune cells [18]. A recent study presented that breast cancers also contained an abundance of tumor-associated macrophages (36.3%), plasma cells (17.9%) and follicular helper T cells (6.9%), whereas eosinophils, monocytes and resting natural killer (NK) cells were scarce [6]. The tumor protective or promoting role of some of these immune cell types is ambiguous. In general, CD4^+^ T helper 1 (Th1) cells, CD8^+^ cytotoxic T cells, NK cells, M1 macrophages, and dendritic cells are considered to be protective against tumor growth, whereas CD4^+^ FOXP3^+^ (forkhead box P3), CD4^+^ Th2 cells, M2 macrophages, and myeloid-derived suppressor cells (MDSCs) promote tumor growth [19,20]. 

Based on the spatial distribution of immune cells in the tumor microenvironment, tumors can be classified as immune-inflamed (also named “hot tumors”), immune-excluded and immune-desert phenotypes (described as “cold tumors”). Tumors with an inflamed phenotype tend to be more responsive to immune checkpoint inhibitors. Accordingly, serious efforts are made to develop strategies that promote the transformation of “cold tumors” into “hot tumors” [21]. 

A study examining 33 diverse cancers from the TCGA database have identified six distinct immune subtypes. They found that among breast cancers (*n* = 944) none was identified as immunologically silent [16].

In summary, there is a high relevance for studies analyzing the complexity of interactions between tumor cells and the tumor microenvironment, leading to a better understanding of immune related changes in different breast carcinoma subtypes and tumor progression. In our study, we (1) compare different breast carcinoma subtypes based on the composition and amount of immune cells detected at different localizations; (2) describe the composition and localization of immune cells in different breast carcinoma subtypes as well as assess their prognostic value; and (3) explore the immune gene expression profile of metastatic and non-metastatic primary breast carcinoma subtypes, to obtain further insight into the prognostic and functional role of immune-related genes in the subsequent metastasis development.

## 2. Results

### 2.1. Systematically Evaluated Spatial Distribution of TILs in Breast Cancer

#### 2.1.1. Distribution of TILs at Different Localizations and Relationship with Subtype

Baseline clinicopathological characteristics of the patients with primary operable breast carcinoma cases (*n* = 309) are shown in Table 1.

To define patient groups based on TIL localization with comparable sizes, we used the thresholds indicated in Figure 1A with vertical lines. Out of the 309 cases, 110 (35.59%) had low amount of sTIL (<5%), and 199 (64.41%) presented ≥5% sTIL. Some degree of peritumoral lymphocytic infiltration (pTIL) was seen in all tumors: score 1 in 225 (72.81%), score 2 in 65 (21.03%) and score 3 in 19 cases (6.14%). Furthermore, 49.19% of the cases presented <2 iTIL count and 50.80% of the cases ≥ 2 iTIL count. 

By comparing the distribution of TILs in different subtypes, we have observed that LUMA exhibits the lowest level of TILs at peritumoral and stromal localization, followed by LUMB1 also showing a significantly lower TIL amount at the peritumoral and stromal region. Regarding the intratumoral region, the TNBC subtype stands out due to its significantly higher iTIL count compared to LUMA, LUMB1 and HER2. The TIL distributions are visualized in Figure 1B. 

#### 2.1.2. TIL Localization According to Clinicopathological Characteristics

Grade 3 tumors show significantly higher levels of pTIL, sTIL and iTIL than either grade 1 (*p* < 0.001) or grade 2 (*p* < 0.001) tumors (Appendix A). According to the size of the tumors, pT1 tumors show significantly lower peritumoral and intratumoral TIL levels than pT2 tumors (Wilcoxon-test *p* = 0.008 and *p* = 0.002, with global Kruskal–Wallis *p*-values of 0.043 and 0.018, respectively), whereas only borderline significant differences in the number of stromal TILs in different pT groups was observed (global Kruskal–Wallis *p* = 0.083, with a Wilcoxon-test *p*-value of 0.042 between pT1 and pT2 tumors) (Appendix A).

By analyzing pN status, only the pN0 vs. pN1 comparison shows a borderline significant difference (*p* = 0.047) in the amount of peritumoral TILs, but the global Kruskal–Wallis p-value is not significant (*p* = 0.25). None of the pairwise comparisons show a significant difference in the amount of stromal and intratumoral TILs in different pN groups (Appendix A).

#### 2.1.3. Prognostic Role of TIL Amount and Localization

Kaplan–Meier analysis resulted in no significant association between distant metastasis-free survival (DMFS) and TIL levels at different localizations (Figure 2A). Multivariate survival analysis also demonstrated the lack of prognostic relevance of TIL levels measured at different localizations (Figure 2B) besides the prominent effect of pN status and subtype on survival. 

Additionally, we also grouped the tumors based on their immune category (hot, cold, and immune excluded). This categorization resulted in 204 cold tumors, 56 hot tumors and 49 immune-excluded tumors. There is a very slight survival benefit for hot tumors vs. cold tumors in a univariate setting, but this trend could not be verified as significant. Immune-excluded tumors behave very similarly to cold tumors in terms of DMFS probability (Figure 2C). Fitting a multivariate Cox-regression model similar to that of Figure 2B, but using the distinct immune categories instead of actual TIL levels at different localizations shows the lack of prognostic significance of the immune category of the tumor (Figure 2D).

When analyzing HR-positive and HR-negative breast carcinoma cases separately, no significant association between DMFS and TIL levels could be either established at any of the localizations (Figure 2A,B).

### 2.2. Immunohistochemical Analysis to Investigate CD4^+^ and CD8^+^ T Cell Subsets Infiltrating Breast Cancer

#### 2.2.1. Spatial Distribution of CD4^+^, CD8^+^ Cells and CD4^+^/CD8^+^ Ratio

Highly heterogenous CD4^+^ and CD8^+^ distributions were observed across the 75 analyzed cases, as well as the amount of CD4^+^ and CD8^+^ T cells varied widely in all the four localizations. Representative images are shown in Figure 3A, and raw data of CD4^+^ and CD8^+^ expression at different tumor localizations in Appendix A.

Given that the amount of CD4^+^ and/or CD8^+^ lymphocytes was measured in different units at different localizations, only CD4^+^/CD8^+^ ratios were compared between tumor regions (Figure 3B). The comparison resulted in a significant global Kruskal–Wallis *p*-value of 0.0078 with a significantly lower CD4^+^/CD8^+^ ratio measured at intratumoral localization than in either of the other three regions (Wilcoxon *p* = 0.0043 compared with peritumoral, *p* = 0.0053 compared with stromal external and *p* = 0.0047 compared with stromal internal regions).

#### 2.2.2. Distribution of CD4^+^, CD8^+^ T Cells and CD4^+^/CD8^+^ Ratio in Breast Carcinoma Subtypes

There was no significant association between the presence of CD4^+^ or CD8^+^ T cells and breast cancer subtype, nor between the CD4^+^/CD8^+^ ratio and any subtype. However, very slight tendencies were apparent. Of note, TNBC and HER2 tumors generally had higher levels of CD4^+^ and CD8^+^ T cells at peritumoral and stromal external level than other subtypes. LUMB2 tumors had high CD4^+^ cell amount at the peritumoral region, as well as higher CD4^+^/CD8^+^ ratios than other subtypes at peritumoral and stromal external localizations. On the other hand, HER2 tumors had higher CD4^+^/CD8^+^ ratios at stromal internal and intratumoral localizations (Figure 3A–C). 

#### 2.2.3. Prognostic Role of CD4^+^/CD8^+^ Ratio

Given that no obvious choice is available for the threshold value, we categorized patients as having “low” levels of CD4^+^ and CD8^+^ T cells and CD4^+^/CD8^+^ ratio if they were below the median value, and as having “high” levels of CD4^+^ and CD8^+^ T and CD4^+^/CD8^+^ ratio if they were above the median value. Thus, the threshold was chosen dynamically based on the available data.

Interestingly, a tendency of patients with lower CD4^+^/CD8^+^ ratios having longer distant metastases-free survival was present. This was only significant for stromal internal ratios (log-rank test *p* = 0.042) (Figure 4A). Analyzing only HR-negative cases did not result in any obvious tendencies between the ratio of CD4^+^/CD8^+^ and DMFS (Figure 4B). 

### 2.3. Immune Gene Expression Differences in Metastatic vs. Non-metastatic LUMA, LUMB1 and TNBC Subtypes

First, we compared peritumoral, stromal and intratumoral TIL assessed on HE stained slides in non-metastatic (Non-met) and metastatic (Met) LUMA, LUMB1 and TNBC cases. Cases were considered as Non-met only if no distant metastases were diagnosed during a minimum of five-year follow up. In total in the three subtypes, 152 Non-met and 51 Met samples were compared. By comparing the peritumoral, stromal and intratumoral TIL amounts in Mets vs. Non-met cases separately for each subtype, no significant differences were observed regarding TIL amounts at any localizations (Appendix A). Albeit not significantly, but a lower sTIL was detected in Met LUMA and LUMB1 cases compared to Non-met ones.

Further investigations of the immune gene assembly of carcinomas diagnosed with distant metastasis vs. carcinomas showing no progression was performed using an immune-oncology gene expression profiling panel provided by NanoString. Using strict filtering criteria based on the length of follow-up (see Methods for further details), 35 cases were selected from the three above mentioned subtypes. Samples were categorized into control (non-metastatic, Ctrl) and metastatic (Met), followed by the quantitative measurement of 730 immune genes in each sample and performing differential expression analysis to detect genes with altered expression between Met and Ctrl. 

In total, 167, 15 and 15 genes showed differential expression in LUMA, LUMB1 and TNBC subtypes, respectively. The luminal subtypes showed a substantially higher number of downregulated genes in Met samples. This was most prominent in LUMA cases where out of the 167 genes, 143 (85.63%) were downregulated and only 24 (14.37%) were upregulated in Met. We observed coordinated downregulation of several interleukins, CT-antigens, cytokines and chemotactic ligands/receptors with some of them known to mediate migration of different immune cell types in the peripheral tissues (Figure 4A). In LUMB1 tumors, 12 out of 15 (80.00%) were downregulated and only 3 (20.00%) were upregulated (*VEGFA*, *PYCARD* and *NOS2A*) in Met cases (Figure 4B). Regarding TNBC samples, the distribution of up- and downregulation was more balanced, with seven genes being downregulated (46.67%) and eight genes being upregulated (53.33%) in Met samples (Figure 4C). 

The differential expression analysis results for all 730 genes and overrepresentation analyses are presented in Appendix A.

Additionally, an unsupervised hierarchical clustering of all 730 immune genes was performed to supplement the differential expression results (Figure 5). The heatmap shows that in case of luminal disease (LUMA and LUMB1 cases), Ctrl and Met samples tend to highly differ from each other, but they have similar gene expression profiles within Met vs. Ctrl groups, regardless of subtype. On the other hand, both Ctrl and Met TNBC cases form a distinct cluster. K-means clustering of the genes resulted in four clusters with diverse attributes. Cluster 1 shows a tendency for upregulation in Ctrl luminal as well as Ctrl and Met TNBC samples. Genes involved in T/B-cell functions, cytotoxicity and regulation are enriched in this cluster (one-sided Fisher’s exact test *p* < 0.001, *p* = 0.001, *p* = 0.003 and *p* = 0.018, respectively). Cluster 2 can be characterized by a high ratio of CT-antigens, interleukins and genes with NK cell functions (one-sided Fisher’s exact test *p* < 0.001, *p* < 0.001 and *p* = 0.033, respectively), and this group of genes exhibits increased expression in Ctrl luminal cases. In cluster 3, there is a weak enrichment for senescence-related genes (one-sided Fisher’s exact test *p* = 0.068), and can be characterized by genes with higher expression in the Met luminal samples. Lastly, cluster 4 shows an upregulation in TNBC samples, and representatives of immune genes with transporter, cell adhesion and cell cycle regulation function are overrepresented (one-sided Fisher’s exact test *p* = 0.002, *p* = 0.060 and *p* = 0.073, respectively). The cluster assignment for each gene and the overrepresentation analysis results can be found in Appendix A.

## 3. Discussion

Tumor immune microenvironment in breast carcinomas is very heterogenous with many aspects not yet identified [10,22,23]. Multiple studies suggest the important role of lymphocytic infiltration in the stroma of breast carcinomas and in the epithelial compartments, but only some studies outline the importance of the accumulation of immune cells in the peritumoral region. The mechanisms leading to the accumulation of immune cells in different tumor compartments are still under considerable debate [10,24,25,26].

By analyzing the peritumoral, stromal and intratumoral compartment separately for lymphocytic infiltration we have found that as a general result, all breast cancer subtypes have tumors with low, intermediate or high TIL infiltrate, but HR negative subtypes show higher immune cell infiltration in all the analyzed localizations compared to LUMA and LUMB1 subtypes. The least immune-infiltrated subtype was LUMA. A recent study analyzing TIL in 987 patients with early ER positive/HER2 negative breast carcinoma cases, also described a low TIL count (median TIL count was 2%) [27].

Comprehensive characterization of the immunological aspects of the tumor microenvironment in HR positive tumors is an unmet need, with the aim of rendering those subtypes presenting lower immunity also susceptible for immunotherapies. There are currently some ongoing clinical trials assessing the combination of immune checkpoint inhibitors with different therapies in HR positive breast cancers [17]. 

Based on the accumulation of immune cells in peritumoral, stromal and intratumoral localization we have categorized our cases into hot, cold and immune-excluded cases according to a recently published study of Kather et al. and we have found very heterogenous immune infiltration patterns in different tumor compartments across the tumors [28]. Only 18.12% of our cases were identified as hot tumors proving that immune activity is generally low in breast cancers. Moreover, survival analysis on our data revealed only a slight tendency of survival benefit for hot tumors vs. cold tumors, which could not be verified in a multivariate setting. There is accumulating evidence that patients who do not respond to immunotherapy present tumors that either have reduced T cells or have T cells located around the tumor (i.e., are immune-excluded) [29,30]. The important question arising from these results is the differences or similarities between the possible mechanisms associated with T cell motility and migration in different tumor compartments. Among the environmental factors that may govern T cell migration to intratumoral regions, the physical structures/barriers of the tissue provided mainly by extracellular matrix structures, and the cellular composition of the tissue highly involved in secreting different factors, such as chemokines, are especially considered to lead to different T cell motility [24,31]. Peranzoni et al. have found that macrophages mediate lymphocytes motility by forming long lasting interactions with CD8^+^ T cells [24]. 

Although the lymphocyte phenotype can also dictate the clinical outcome, only a limited number of studies have investigated the importance of subsets of TILs at different localizations. CD4^+^ and CD8^+^ T cells are the main types of lymphocytes in breast cancers and play a central role in the induction of efficient immune responses against tumors. While the majority of cancer immunotherapies focus on CD8^+^ T cells, considered as the key players in tumor defense, the potential role of CD4^+^ T helper cells has remained mostly unexamined, even if it is becoming clear that CD4^+^ T cells play a critical role in developing and sustaining effective anti-tumor immunity [32]. The significance of the CD4^+^/CD8^+^ ratio has been explored in a high number of tumor types resulting in very different prognostic significance [9,33,34,35,36]. In our study, we have observed a wide range of CD4^+^ and CD8^+^ immune cell expression in all localizations. Albeit significant only for one tumor region, namely, for the stromal internal region, we detected a general tendency for cases with a lower CD4^+^/CD8^+^ ratio exhibiting longer DMFS, indicating that higher CD8^+^ T cell accumulation is favorable for the patients. 

Growing evidence suggests that for a better clinical response to immunotherapies, the accumulation of CD8^+^ T cells in the stroma is not sufficient and the inability of CD8^+^ T cells to reach and contact tumor cells is an important mechanism of resistance to cancer immunotherapy [24,30,37]. More interesting is the observation based on a large independent cohort of breast cancer patients, which was that patients presenting with accumulation of high numbers of CD8^+^ T cells in tumor stroma had shorter overall survival [38]. This is partly explained by the observation that the highest accumulation of CD8^+^ cells in the tumor stroma is associated with elevated levels of IL-17 producing immune cells as well as with a higher number of neutrophils considered as having pro-tumorigenic activity [39,40]. The controversies around the prognostic value of CD8^+^ T cells are partly explained by recent technological advances, providing important insights into the heterogeneity of CD8^+^ TILs. CD8^+^ T cells have multiple ways to eliminate tumors, and they can directly target cancer cells or indirectly target tumor stromal cells. Which mechanism is relevant in different tumor types, is a future research challenge [40].

Despite its importance, our understanding of immune-related gene expression in breast carcinomas, and especially in different breast carcinoma subtypes, is still rather limited. We detected significant differences in immune genes expression in metastatic (Met) vs. non-metastatic (Ctrl) primary breast carcinomas of different subtypes. It is partly documented that when investigating sample pairs of primary and distant metastatic cancers, most of the immune cell types and immune functions are depleted in metastases compared to the corresponding primary breast carcinomas [41]. A very recent study described reduced expression of immunity related genes in lymph node metastases of luminal breast cases when compared to primary breast carcinomas [5]. It is questionable how early the depletion of immune cells occurs during the steps of tumor progression. In our study, as a general tendency, LUMA and LUMB1 primary BCs diagnosed later with distant metastases showed lower expression of immune genes compared to the cases where no distant metastases occurred during the long follow-up period. To be exact, 85.63% and 80.00% of the differentially expressed genes between Ctrl and Met in the luminal subtypes showed overexpression in Ctrl cases. Important to note that some of these genes are, in general, low in abundance indicating that we observed subtle differences that might be difficult to capture at the protein level by immunohistochemistry. In contrast, for TNBC cases the number of up- and downregulated genes was fairly balanced (46.67% was detected as overexpressed in Ctrl). 

The Met LUMA vs. Ctrl LUMA showed the largest differences in terms of differentially expressed genes, presenting a list with 167 genes, compared to 15–15 genes detected with altered expression across LUMB1 and TNBC subtypes. There was no notable overlap between the differentially expressed genes across the luminal vs. TNBC cases; however, certain CT antigens (such as members of the MAGE family in luminal subtypes or CTAG1B gene in TNBC) showed downregulation in Met samples of all subtypes. Considering general trends in the data, Met and Ctrl luminal subtypes exhibited high similarity, with CT-antigens, interleukins and genes with NK cell functions, showing a more increased expression in both Ctrl luminal subtypes. On the other hand, TNBC was yet again a distinct group showing less similarity with the luminal cases. Further studies are needed as the exact role of some of these genes and gene categories is less known in breast carcinomas.

The immunological differences identified between and among the different subtypes underline the importance of exploring distinct immunotherapy modalities or other therapeutical strategies in these three molecular subtypes. The non-exhaustive list of genes being upregulated in the Met vs. Ctrl comparison in the three subtypes, namely, *IL17A*, *IL17RB*, *IL6ST* and the *TNF* superfamily; *STAT3* and *STAT6* in LUMA; *VEGFA*, *PYCARD* and *NOS2A* in LUMB1; and *ATG5*, *DUSP4*, *GATA3*, *CEACAM6*, *AKT3* and *BCL2L1* in the TNBC subtype, also supports a previous suggestion of targeting other molecules and pathways in different breast carcinoma subtypes [42,43]. Understanding these major differences in immune cell composition of different breast cancer subtypes may pave the way to the development of immune-modulation therapies that may delay or prevent metastatic progression.

We acknowledge that our study has certain limitations. Although the study cohort spans over 14 years where the majority of the cases are present with a long follow-up period, we cannot account for eventual differences in therapeutic protocols during this long time interval. The cases needed to be further categorized based on subtype and important clinicopathological characteristics, such as grade or pN, resulting in a smaller number of cases for some statistical analyses.

## 4. Materials and Methods

### 4.1. Patients and Clinicopathological Characteristics

Our initial cohort consisted of 487 patients. The following cases were excluded from the initial cohort: local recurrences, patients who underwent neoadjuvant therapy, patients simultaneously diagnosed with distant metastases at first diagnosis, and cases with incomplete/missing follow-up data. Finally, our retrospective study assembled data from 309 patients diagnosed with invasive breast cancer at Semmelweis University, 2nd Department of Pathology between 2000 and 2014. The mean age of the patients at diagnosis was 58.1 years (range 27–92 years).

Clinicopathological data of the patients were obtained from the files of Semmelweis University, 2nd Department of Pathology and from the Semmelweis University Health Care Database with the permission of the Hungarian Medical Research Council (ETT-TUKEB 14383/2017). Distant metastasis-free survival (DMFS) was defined as the time from the date of primary breast cancer diagnosis to the occurrence of first distant metastasis and recurrence-free survival (RFS) as the time from the primary breast cancer diagnosis to the occurrence of any breast cancer related disease: local recurrence, second primary breast carcinoma, metastases in the ipsilateral lymph nodes or any distant metastasis. All patients were followed-up until the date of death or until January 30, 2021 (median follow-up time was 99.1 months for DMFS and 84.8 months for RFS

The following data were taken into consideration (Table 1): age at diagnosis, Nottingham grade, pathologic tumor size (pT), nodal involvement (pN), surrogate breast carcinoma subtype as defined based on four (estrogen receptor (ER), progesterone receptor (PR), Ki67 index (marker of proliferation) and HER2) immunohistochemical markers and according to the 2013 St. Gallen Consensus Conference recommendations. Luminal A (LUMA) tumors are defined as ER and PR positive, HER2 negative, Ki-67 “low” (Ki-67 < 20%) tumors, Luminal B-HER2 negative (LUMB1) tumors as ER positive, HER2 negative and Ki-67 “high” (≥20%) and/or PR “negative or low” (PR cut-point = 20%), Luminal B-HER2 positive (LUMB2) as ER positive and HER2 overexpressed or amplified and HR negative, and triple negative breast carcinomas (TNBC) as HR and HER2 negative [44].

ER, PR, HER2 status and Ki67 index were evaluated by immunohistochemistry (IHC). All immunohistochemical stains were carried out and routinely evaluated at the Semmelweis University, 2nd Dept. of Pathology, Hungary. Cut-off values for ER and PR status were 1% of tumor cells with nuclear staining and defined as positive or negative [45]. HER2 status was determined either as protein overexpression or *HER2* gene amplification detected by fluorescence in situ hybridization (FISH) [46,47,48]. 

### 4.2. TIL Assessment

For TIL assessment the HE stained slides were scanned with Pannoramic 1000 scanner (3DHistech, Budapest, Hungary) and evaluated in peritumoral, stromal and intratumoral regions. TILs immediately adjacent to the invasive margin were defined as peritumoral TIL (pTIL) and their abundance was scored on a semiquantitative scale as follows: score 0, no immune cells at the tumor’s margin; score 1, mild and patchy aggregates of immune cells; score 2, presence of prominent band-like immune cells infiltration; and score 3, very prominent, florid cup-like immune cells infiltrate [49,50].

The number of stromal TILs (sTIL) was measured in %, based on the guidelines of TIL assessment in solid tumors, provided by the International Immuno-Oncology Biomarker Working Group. Accordingly, sTIL was defined as area occupied by mononuclear inflammatory cells over total stromal area [49].

Lymphocytes in contact with or within the tumor epithelium were defined as intratumoral (iTIL) [49]. Due to the low percentage of immune cells in the intratumoral area, lymphocytes were counted individually. The mean of at least four counts from four different regions of the tumor was calculated and recorded for intratumoral area by using ×400 magnification.

Based on the suggested classification of Kather et al. (2018) the tumors were also grouped into inflamed (“hot”), non-inflamed (“cold”) and “immune excluded” categories [28]. High lymphocytic density outside of the tumor with a low density inside the tumor can be described as “immune excluded”. Low density inside and outside is “cold” and high density inside the tumor is “hot” regardless of cell density outside of the tumor [28]. As a cut-off value for high vs. low cell density, we used the median cell density for each localization.

### 4.3. Immunohistochemical Analysis to Investigate CD4^+^ and CD8^+^ T Cell Levels

By using Bond Automated Immunostainer (Leica Microsystems, IL, USA), dual immunostaining of CD4:CD8 was performed on 75 formalin-fixed paraffin-embedded (FFPE) breast carcinoma cases with mean sTIL levels above 1%. Anti-CD4 (SP35) Ventana Rabbit monoclonal and anti-CD8 (SP57) Ventana Rabbit monoclonal antibodies were used in the study. The optimization of each antibody with each chromogen and antibody staining order was set before the staining. Finally, 3,3’-Diaminobenzidine (DAB) chromogen was used for CD4^+^ T cells visualization followed by CD8^+^ T cells immunostaining where the chromogen was alkaline phosphatase. The clinicopathological data of the 75 cases are presented in Table 1. 

After reviewing the immunostaining of the 75 cases, we observed a very heterogeneous staining of CD4^+^ and CD8^+^ T cells in different tumor regions. Therefore, the CD4^+^ and CD8^+^ cell quantification was performed in the following regions: peritumoral, stromal external (invasive margin), stromal internal (central tumor) and intratumoral areas. The definitions of “invasive margin” and “central tumor area” were based on the recommendations of Hendry S et al. (2017). The invasive margin is defined as a 1 mm region centered on the tumor border separating the malignant cell nests from the host tissue. The central tumor represents the remaining tumor area [49].

Each tumor was divided into four parts. The percentages of CD4^+^ and CD8^+^ T cells were assessed in the peritumoral area and in the external and internal stroma, whereas in four intratumoral areas the CD4^+^ and CD8^+^ lymphocytes were counted. The mean values of the four parts in each area (peritumoral, stromal external, stromal internal and intratumoral) was taken into consideration for statistical analyses. The median value of CD4+ and CD8+ T cell amount in the various tumor compartments is listed in Appendix A. 

### 4.4. Statistical Analyses for TIL and CD4^+^/CD8^+^ Asessment

Statistical analyses were performed using R version 4.0.4. For visualizations, the ggplot2 3.3.3, gridExtra 2.3, cowplot 1.1.1, survminer 0.4.8, survival 3.2-7 and forestmodel 0.6.2 R packages were used.

Prognostic relevance of TIL levels at different localizations was assessed in a two-step manner. First a univariate analysis was performed with Kaplan–Meier estimates of the survival curves using log-rank tests to compare the survival of patients stratified into two categories based on the given threshold for the specific TIL localization. Thresholds were chosen as the median value of the measured quantity (score 1 or lower vs. above score 1 for peritumoral TIL; lower than 5% vs. 5% or higher for stromal TIL; below 2 vs. 2 or higher for intratumoral TIL). Following univariate analysis, a multivariate Cox-regression model was fitted to the available data using covariates of grade, nodal involvement, and tumor subtype besides TIL levels at the above localizations. Both stromal and intratumoral TIL were treated as continuous variables for this analysis. The proportional hazards assumption was verified by calculating the correlation coefficient between transformed survival time and the scaled Schoenfeld residuals using cox.zph function of the survival R package, and checking if any of the correlations were significant. If a given variable resulted in a significant *p*-value, patients were stratified based on its value in the final multivariate model, and different baseline hazards were used for different patient groups. Final hazard ratios are presented as the mean values of the results obtained by stratification.

Association between TIL levels and clinicopathological parameters was assessed with pairwise comparisons using Wilcoxon tests and a global comparison of all patient groups using Kruskal–Wallis tests. *p*-values below 0.05 were considered significant.

The amount of CD4^+^ and CD8^+^ T cells, as well as the CD4^+^/CD8^+^ ratio was separately tested in statistical analyses for each of the four tumor regions (peritumoral, stromal external, stromal internal, intratumoral). Given that intratumoral CD4^+^ and CD8^+^ T cell levels are defined on a different scale than the T cell levels at the rest of the localizations, we refrained from directly comparing these data. CD4^+^/CD8^+^ ratios, on the other hand, are devoid of any artificial units of the data and can be compared across different localizations. This comparison was performed in a pairwise manner between different tumor regions using Wilcoxon tests and a global Kruskal–Wallis test. Comparison between different subtypes was carried out as discussed above. Survival analysis was performed in the same way as for TIL levels. 

### 4.5. Patient Selection for Immune Gene Expression Analysis

FFPE tissue blocks of non-metastatic control (Ctrl) and metastatic (Met) primary breast carcinomas of LUMA (6 Ctrl and 6 Met), LUMB1 (5 Ctrl and 6 Met) and TNBC (6 Ctrl and 6 Met) cases were selected from the previously presented 309 cases. To compare cases of different subtypes, we decided to omit HER2-positive cases (LUMB2 and HER2+ subtypes) receiving HER2 targeted therapies. Follow-up time in the Ctrl group was important criteria in case selection. Accordingly, the median follow-up time was 141.83 months in Ctrl LUMA group, 127.80 months in Ctrl LUMB1 group and 140.17 months in Ctrl TNBC group (Table 2). 

### 4.6. RNA Extraction and NanoString nCounter Analysis

Tumor cellularity was assessed prior to RNA isolation on HE stained slides and ranged between 60% and 90%. RNA was extracted from three-to-five 10 μm FFPE curls of whole sections using the QIAGEN RNeasy FFPE Kit (Qiagen, Venlo, Denmark) and quantified with Qubit Fluorometer (Invitrogen, Waltham, MA, USA). The RNA samples were diluted to 50 ng/µL. 

The NanoString *nCounter PanCancer Immune Profiling Panel* is a unique 770-plex gene expression panel containing 730 immune-related genes and 40 housekeeping genes (https://www.nanostring.com/products/ncounter-assays-panels/oncology/pancancer-immune-profiling) (accessed on: 14 April 2021) [51].

According to the manufacturer’s guide, 8 µL of Master Mix (Mixture of Reporter CodeSet and Hybridization Buffer) was added to 5 µL of sample RNA in a tube. After adding 2 µL of Capture ProbeSet to each tube, the solution was gently mixed, briefly spun and placed immediately in a pre-heated 65 °C thermal cycler for 24–26 h. After incubation, the samples were immediately placed into the nCounter Prep station, and then analyzed in the Digital Analyzer (nCounter FLEX Analysis System, NanoString, Seattle, WA, USA). Measurements were taken at high sensitivity with 555 FOV.

### 4.7. NanoString Data Processing and Statistical Analyses

The data processing and statistical analysis was performed in R version 4.0.4. For visualizations, the ggplot2 3.3.3, gridExtra 2.3, cowplot 1.1.1 and ComplexHeatmap 2.6.2 R packages were used.

Quality control and data normalization was performed based on the guidelines provided in Bhattacharya et al. as their normalization pipeline was shown to remove technical variation more robustly than the traditional workflow provided by nSolver [52]. The process consisted of 3 major steps: (1) Technical quality control (QC) of the samples, (2) Selection of appropriate housekeeping genes for normalization, and (3) Normalization using RUVSeq (Remove Unwanted Variation from RNA-Seq Data) method.

All samples passed the technical QC (i.e., no flags were present for Imaging, Binding Density, Positive Control Linearity and Limit of Detection). The suitable housekeeping genes were then selected based on the following criteria: (i) No housekeeping gene count values should be below the mean count values of the negative control probes, (ii) No differential expression between Ctrl and Met in either subtype (assessed by performing negative binomial regression analysis on the raw counts), (iii) Similar mean and similar coefficient of variation (where outlier values were identified using the 1.5xIQR rule), and (iv) Good correlation (Spearman’s correlation coefficient > 0.49) with other housekeeping genes. In total the following 30 housekeeping genes were used for the normalization: *DNAJC14*, *DHX16*, *ZNF143*, *HDAC3*, *CNOT10*, *SAP130*, *AGK*, *POLR2A*, *AMMECR1L*, *SF3A3*, *COG7*, *TMUB2*, *ZC3H14*, *DDX50*, *G6PD*, *ALAS1*, *PPIA*, *SDHA*, *TBP*, *ZNF346*, *MTMR14*, *ERCC3*, *EIF2B4*, *TLK2*, *TRIM39*, *USP39*, *PRPF38A*, *GPATCH3*, *CNOT4*, and *HPRT1*. The first step of the normalization was an upper-quartile normalization [53] followed by the estimation of one dimension of unwanted variation with the RUVr function of the RUVSeq 1.24.0 R package [54]. The DESeq2 1.30.1 R package was then used to compute a variance stabilizing transformation of the original count data [55], and finally the unwanted variation was removed with the removeBatchEffect function from limma 3.46.0 R package [56]. We conducted differential expression analysis using DESeq2 1.30.1 R package. Met LUMA vs. Ctrl LUMA, Met LUMB1 vs. Ctrl LUMB1 and Met TNBC vs. Ctrl TNBC comparisons were made. Significance level was set to 0.05 and *p*-values were adjusted for multiple testing with the Benjamini–Hochberg method.

Gene annotations were accessed by downloading the “nCounter Human Pancancer Immune Profiling Panel Gene List” supporting document from the NanoString website (https://www.nanostring.com/products/ncounter-assays-panels/oncology/pancancer-immune-profiling) (accessed on: 14 April 2021). On the “Annotations” sheet, the “Gene Class” and “Immune Response Category” for each gene was extracted (see this information also in Appendix A). Overrepresentation analysis [57] was performed via a one-sided Fisher’s exact test to assess whether certain Immune Response Categories are enriched in a list of differentially expressed genes.

Unsupervised clustering of the 730 gene expression values (after normalization and variance stabilizing transformation) was completed using the Heatmap function from the ComplexHeatmap R package. The following settings were used: k-means clustering with 4 clusters (row_km = 1 and row_km_repeats = 1), Euclidean distance and average linkage. Overrepresentation analysis for the cluster elements was performed in the same manner as discussed previously.

## 5. Conclusions

In summary, we showed that the extent of immune infiltration at different tumor localizations differs between subtypes, with LUMA showing lower TIL levels at all localizations, and LUMB1 at peritumoral and stromal regions. Most analyzed tumors belonged to the immunologically cold category, providing further evidence that immune activity is generally low in breast cancers. However, our data indicate that a low level of TILs does not unequivocally equate to disease progression. TIL levels correlated with tumor grade and size, but not with lymph node involvement. CD4^+^/CD8^+^ ratios were lower in the intratumoral region compared to peritumoral and stromal parts. The number of TILs at different localizations was not found prognostic for DMFS, but our data indicated the prognostic significance of the intratumoral CD4^+^/CD8^+^ ratio. Differences in immune gene expression observed between metastasizing and non-metastasizing breast carcinomas call attention to the important role of altered immune function in breast cancer progression.

## Figures and Tables

**Figure 1 cancers-14-01942-f001:**
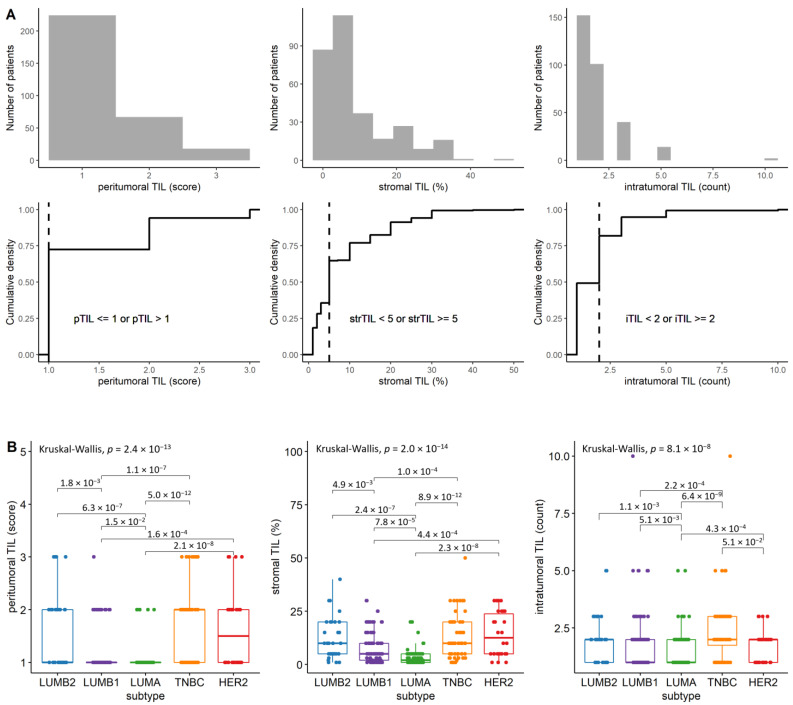
The distribution of TIL categories across the analyzed cases: (**A**), The distribution of patients (*n* = 309) in the different TIL localization categories. Patients were stratified based on the thresholds indicated with vertical lines for subsequent analyses. (**B**), The distribution of peritumoral, stromal and intratumoral TILs in different breast carcinoma subtypes (*n* = 305). Only pairwise comparisons with Wilcoxon-test *p*-values less than 0.100 are shown.

**Figure 2 cancers-14-01942-f002:**
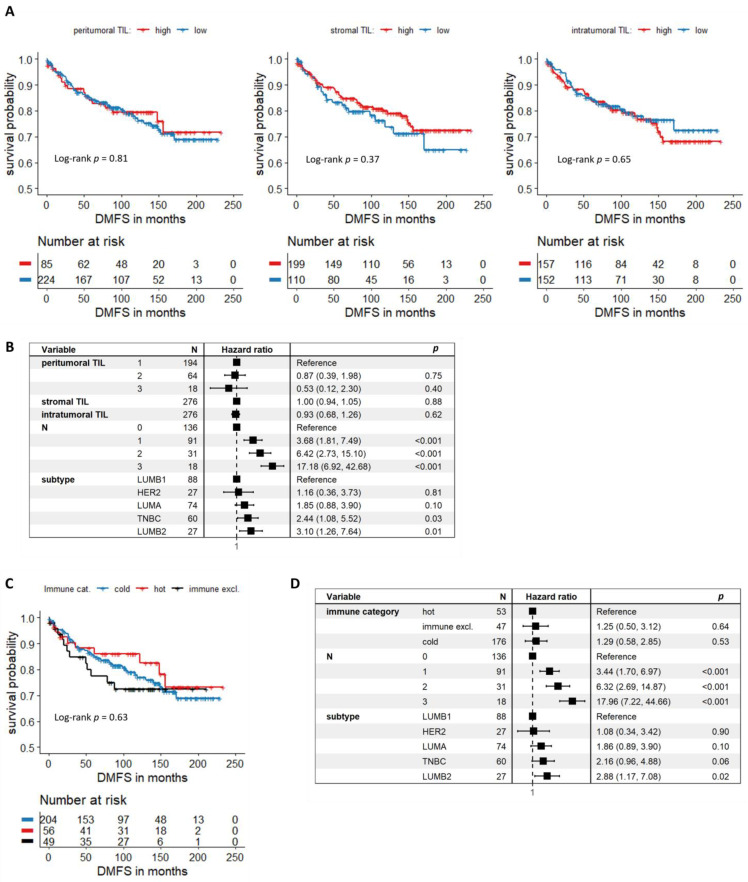
Associations between DMFS and TIL levels at different localizations: (**A**), Kaplan–Meier curves of patients with high and low pTIL, sTIL and iTIL (*n* = 309). (**B**), Forest plot depicting multivariate Cox regression results where TILs at different localizations were treated as independent variables. Patients (*n* = 276) were stratified based on tumor grade. (**C**), Kaplan–Meier curves of patients with cold, hot and immune-excluded tumors (*n* = 309). (**D**), Forest plot depicting multivariate Cox regression results where one independent variable is the tumors’ immune category. Patients (*n* = 276) were stratified based on tumor grade.

**Figure 3 cancers-14-01942-f003:**
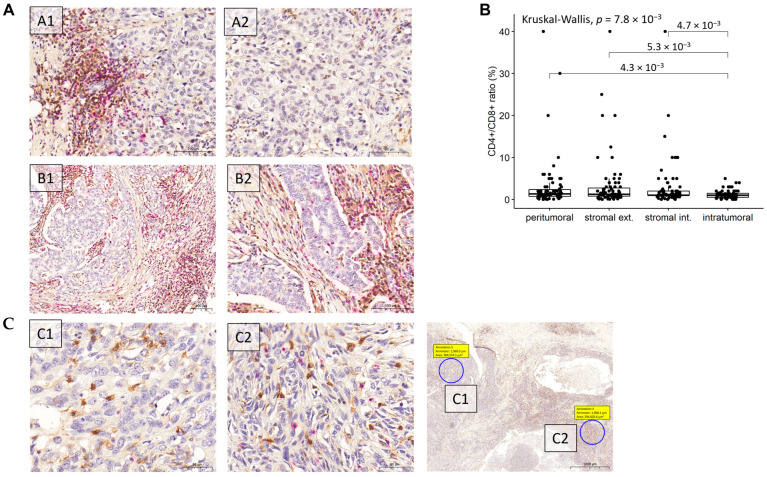
Spatial distribution of CD4+, CD8+ cells: (**A**), Representative IHC images of CD4^+^ (brown) and CD8^+^ (red) immune cells. A1–A2: CD4^+^ and CD8^+^ immunostaining of a HER2+ breast carcinoma subtype of grade 2, pT1c, pN0. High peritumoral CD4^+^ and CD8^+^ expression (A1). Markedly reduced CD4^+^ and CD8^+^ expression in the intratumoral region of the tumor (A2). (**B**), B1–B2: CD4^+^ and CD8^+^ immunostaining of grade 2, pT2, pN1 LUMB1 breast carcinoma case. High peritumoral and stromal CD4^+^ and CD8^+^ immune cells expression (B1). High CD4^+^ and CD8^+^ expression in stromal region, whereas in the intratumoral compartment higher CD8^+^ and reduced CD4^+^ positivity is observed (B2).(**C**), C1–C2: Different intratumoral CD4^+^ and CD8^+^ expression in a grade 3, pT2, pN0 TNBC case. By analyzing two different intratumoral regions (C1 and C2) differences are observed in the presence of CD8^+^ immune cells. (**B**), Comparison of CD4^+^/CD8^+^ ratios at different tumor regions (*n* = 298). Global Kruskal–Wallis and Wilcoxon-tests were performed, and only *p*-values < 0.100 are shown.

**Figure 4 cancers-14-01942-f004:**
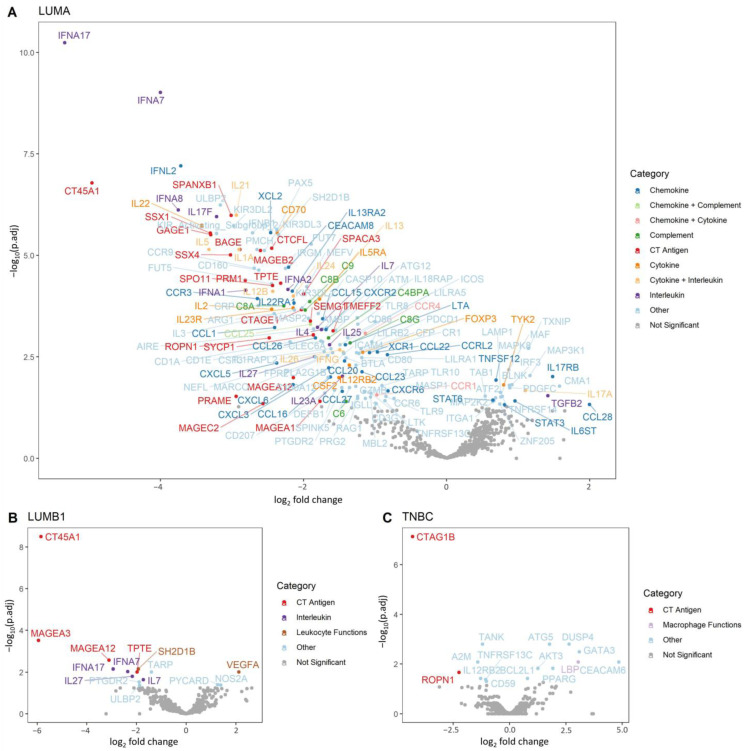
Differential expression analysis of immune genes (*n* = 730) in Met vs. Ctrl comparison for three breast carcinoma subtypes. Results are shown separately for each subtype: (**A**), LUMA, (**B**), LUMB1 and (**C**), TNBC. The *x*-axis shows the log2 fold change between Met and Ctrl, and the *y*-axis shows −log10-transformed adjusted *p*-values from differential expression analysis. Genes that remained significant after Benjamini–Hochberg adjustment (adjusted *p* < 0.05) are colored according to a reduced list of immune response categories (highlighting only immune response categories that showed enrichment among the differentially expressed genes).

**Figure 5 cancers-14-01942-f005:**
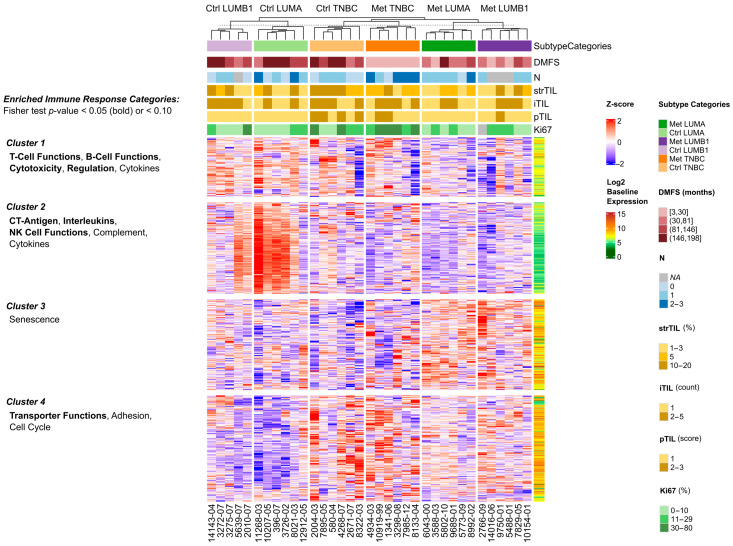
Unsupervised clustering of the 730 immune genes’ normalized expression: The samples (*n* = 35) are grouped according to their subtype categories, and the four gene clusters are annotated with enriched immune response categories (one-sided Fisher’s exact test). The log2 baseline expression of the genes is shown on the right side of the heatmap.

**Table 1 cancers-14-01942-t001:** Clinicopathological characteristics of the breast carcinoma cases analyzed in the study.

Tumor Characteristics	Value	Nr. (%) of Patients for TIL Assessment (309 Cases)	Nr. (%) of Patients for CD4 and CD8 Double Staining (75 Cases)
Age at diagnosis	≤50 year	78 (25.24%)	22 (29.33%)
	>50 year	231 (74.75%)	53 (70.66%)
pT	pT1	148 (47.90%)	32 (42.67%)
	pT2	130 (42.07%)	35 (46.67%)
	pT3	19 (6.15%)	4 (5.33%)
	pT4	12 (3.88%)	4 (5.33%)
pN	pN0	136 (44.01%)	24 (32.00%)
	pN1	91 (29.45%)	27 (36.00%)
	pN2	31 (10.03%)	8 (10.67%)
	pN3	18 (5.83%)	10 (13.33%)
	pNx	33 (10.68%)	6 (8.00%)
Tumor grade	1	62 (20.06%)	9 (12.00%)
	2	131 (42.39%)	22 (29.33%)
	3	116 (37.54%)	44 (58.67%)
Subtype	LUMA	87 (28.16%)	13 (17.33%)
	LUMB1	98 (31.72%)	20 (26.67%)
	LUMB2	30 (9.71%)	8 (10.67%)
	HER2	30 (9.71%)	10 (13.33%)
	TNBC	60 (19.42%)	22 (29.33%)
	No data	4 (1.29%)	2 (2.67%)
Histological type	Invasive ductal carcinoma of no special type (IDC-NST)	271 (87.70%)	71 (94.67%)
	Invasive lobular carcinoma (ILC)	25 (8.09%)	3 (4.00%)
	Other	13 (4.21%)	1 (1.33%)
Recurrences	YES	78 (25.24%)	34 (45.33%)
	NO	227 (73.46%)	41 (54.67%)
	No data	4 (1.29%)	0 (0.00 %)
Distant metastasis	YES	65 (21.04%)	33 (44.00%)
	NO	240 (77.67%)	42 (56.00%)
	No data	4 (1.29%)	0 (0.00%)
pTIL (score)	0	0 (0.00%)	0 (0.00%)
	1	225 (72.82%)	40 (53.33%)
	2	65 (21.04%)	25 (33.33%)
	3	19 (6.15%)	10 (13.33%)
sTIL	<5%	110 (35.60%)	10 (13.33%)
	5–10%	128 (41.42%)	34 (45.33%)
	11–20%	44 (14.24%)	21 (28.00%)
	>20%	27 (8.74%)	10 (13.33%)
iTIL (count)	<2	152 (49.19%)	20 (26.67%)
	≥2	157 (50.81%)	55 (73.33%)

**Table 2 cancers-14-01942-t002:** The clinicopathological data of the 35 cases selected for immune gene expression analyses.

Cases	Subtype Categories	Histological Type	Age at Diagnosis	Grade	pT	pN	sTIL %	iTIL (Count)	pTIL (Score)	Distant Metastases (Yes—1, No—0)	DMFS (Months)
1	Met LUMB1	IDC-NST	43	3	1	1	5	1	1	1	52
2	Ctrl LUMA	IDC-NST	42	1	1	0	10	2	1	0	168
3	Met TNBC	IDC-NST	51	2	2	1	5	2	2	1	18
4	Ctrl LUMA	IDC-NST	77	2	2	2	5	1	1	0	63
5	Ctrl LUMA	IDC-NST	60	2	1	1	3	1	1	0	138
6	Met TNBC	IDC-NST	92	3	4	0	20	3	2	1	15
7	Met LUMB1	IDC-NST	74	2	1	x	1	1	1	1	3
8	Ctrl LUMB1	IDC-NST	30	3	2	0	20	5	1	0	173
9	Ctrl TNBC	IDC-NST	57	3	2	0	20	3	2	0	198
10	Ctrl LUMB1	IDC-NST	49	3	1	0	1	1	1	0	146
11	Ctrl TNBC	IDC-NST	65	3	2	0	5	1	1	0	73
12	Met LUMB1	IDC-NST	56	2	2	1	1	1	1	1	65
13	Ctrl LUMB1	IDC-NST	69	2	1	1	5	2	1	0	147
14	Ctrl LUMB1	IDC-NST	59	3	2	1	15	2	1	0	118
15	Met TNBC	IDC-NST	34	3	1	3	3	2	1	1	23
16	Met LUMA	IDC-NST	47	2	1	1	1	1	1	1	8
17	Ctrl LUMA	IDC-NST	54	1	2	0	1	1	1	0	196
18	Ctrl LUMA	IDC-NST	56	2	1	1	1	1	1	0	150
19	Ctrl TNBC	IDC-NST	34	3	2	1	10	1	3	0	148
20	Met TNBC	IDC-NST	66	2	2	3	5	1	1	1	21
21	Met LUMB1	IDC-NST	60	3	2	x	2	2	1	1	7
22	Ctrl LUMB1	IDC-NST	62	1	1	x	2	2	1	0	55
23	Met LUMA	IDC-NST	61	1	1	0	2	1	1	1	33
24	Ctrl TNBC	IDC-NST	64	3	1	1	10	2	1	0	151
25	Met LUMA	IDC-NST	74	2	2	1	5	2	1	1	152
26	Met LUMA	IDC-NST	49	1	1	1	2	1	1	1	40
27	Met LUMB1	IDC-NST	52	2	2	1	15	5	1	1	122
28	Ctrl TNBC	IDC-NST	62	2	1	1	10	3	2	0	137
29	Met TNBC	IDC-NST	52	3	1	3	5	1	1	1	27
30	Ctrl LUMA	IDC-NST	75	3	2	2	15	2	1	0	136
31	Met TNBC	IDC-NST	47	3	2	2	5	2	1	1	9
32	Ctrl TNBC	IDC-NST	64	3	2	0	5	2	2	0	134
33	Met LUMA	IDC-NST	59	1	1	2	5	1	1	1	116
34	Met LUMA	IDC-NST	47	3	2	1	5	2	1	1	81
35	Met LUMB1	IDC-NST	61	1	1	x	15	2	2	1	57

## Data Availability

The authors declare that the data supporting the findings of the presented study are available within the article. Detailed clinical data of individual patients cannot be provided due to ethical restrictions but are available upon reasonable request from the corresponding author. The most important clinicopathological data are presented in Table 1. The script used for the normalization and statistical analysis of the NanoString data is publicly available at https://github.com/bszeitz/Immune_breast_carcinoma (accessed on: 14 April 2021), and the normalized immune gene expression table is provided in Appendix A.

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
