# Peer review of "Investigating the Prognostic Relevance of Tumor Immune Microenvironment and Immune Gene Assembly in Breast Carcinoma Subtypes"

_cancers, 2022, doi:10.3390/cancers14081942_

Round 1

Reviewer 1 Report

The authors present an extensive and really careful analysis of the presence of tumor infiltrating lymphocytes (TIL) in the tumor itself, its stroma and / or tumor environment in relation to the type and stage of the disease as well as the outcome of the patients. The research was performed on a statistically robust number of tumor samples. The results confirm already known information on the relationship between the rate of TIL infiltration and the sensitivity of the tumor to systemic treatment. The fact that the localization of the presence of TIL was not related to the outcome may have been due to the temporal variability (limited timespan) of the infiltration of different sites. Histological specimen should always be considered only as a  snapshot of lymphocyte infiltration distribution at the time of histological processing. Surprisingly, even the amount of TIL did not favourably affected the prognosis. Recently  demonstrated correlation between the CD4 / CD8 ratio and DMFS suggests that the accumulation of CD8 + lymphocytes in the tumor stroma may not always mean a better prognosis of patients, probably due to IL 17 production and neutrophil attraction, which may have a pro-tumor effect. The missing correlation between TIL and the involvement of regional nodes is also of interest. Other conclusions are either not very surprising, such as the correlation of TIL with high tumor grade or or with the size of the primary tumor, or not entirely clear. I think that the reason may be in the heterogeneity between tumor subtypes which have a different biological nature and a completely different prognosis without dependence on their own microenvironment and also the fact that the authors themselves mention the low immunogenicity of breast cancer in general.  The results themselves suggest a likely further path of research, namely the need to dive into the genomic level of the immune response and to detect up- or down-regulation of the relevant genes individually, taking into account the subtype of breast cancer. However, the authors did a great deal of good work. The text is supplemented by graphs, pictures and tables and recent literature is also cited. As part of the continuation of this interesting research, I would recommend focusing on a smaller number of tumor subtypes and examining in detail the possible immunological regulation of metastasis in connection with a change in the involvement of selected key genes.    

Author Response

Thank you for the review and for the very positive and constructive comments.

Reviewer 2 Report

Szeitz et al. conducted and interesting study on the immunological differences of breast carcinoma subtypes.

The study is well planned and the paper is well written with a long and exhaustive introduction which is the right background to the aim of the study.

Methods are clear and the Authors provide all the piece of information useful for the reproducibility.

The results are sound and well presented throughout the text and in the tables; accordingly, the discussion leads the reader to the conclusions which are in line with the scope of the study: microenvironment and immune gene expression drives the difference of metastatic mechanism for hormone receptors positive and triple negative breast cancers.

Author Response

Thank you for the positive evaluation of our study.

Reviewer 3 Report

Dear Authors:

The manuscript by Szeitz et al has demonstrated Differences in immune gene expression observed between metastatizing and non-metastatizing breast carcinomas call attention to the important role of altered immune function in breast cancer progression. Strongly suggest for publishing. Minor linguistic improvement is needed.

Author Response

Thank you for the positive opinion.

Reviewer 4 Report

The authors studied the immunologic landscape of breast cancer, compared the features in between each subtypes, primary and metastatic tumors. They used H&E stain to review the tumor infiltration lymphocytes (TILs) and stratified them by regions: peritumoral, stromal and intratumoral regions. No survival benefit was detected based on the spatial distribution of TILs. They then worked on CD4/CD8 cells; the result was that Lower CD4+/CD8+ ratio at the stromal internal tumor region indicated longer distant metastasis-free survival. Finally, they used Nanostring to investigate the immune gene profile.

Several points were concerned.

  1. The survival of the study cohort was not similar to the survival of the general breast cancer population. In general, luminal A is the best, Her2-positive and TNBC are poor. However, in this cohort, prognosis of Her2 type breast cancer was similar to Luminal B1; Luminal B2 was the worst, significantly worse compared to luminal B1. Luminal A was worse, too. The potential reasons may be different clinical stage, age or other therapeutic issues in between the 5 subtypes. That means that the selection bias existed in this study, and difficult to be corrected by statistics. When the study design was not adequate, the results cannot be believed.
  2. In previous studies, TILs have the prognostic impact of Her2 type and TNBC. However, the authors did not explain why the TIL NOT affect the survival?
  3. The TILs did not correlate the Nanostring gene expression. Less TILs was noted in the luminal subtypes, but high immunogenicity (by Nanostring) was found in the luminal A and B1 in the figure 5. The authors did not explain the association in between TILs (by H&E stain) and Nanostring data. In addition, did authors not perform Nanostring on Luminal B2 and Her2 type breast cancer? The figure 5 did not contain these information.
  4. In the supplementary excel file, the CD4 expression level of Luminal B1, luminal A, TNBC was very similar. Was the amount of CD4 T cell similar in each subtype? The figure S3A showed difference of CD4 T cells in each subtype. The ratio in the figure S3C was very similar in each sample. The author should show the raw data of CD4, CD8 or the ratio of sample.
  5. The resolution of the figure 3 was not enough.

Author Response

Thank you for the comments and questions. Please find attached the responses discussed point-by-point..
